# Design, Assessment and Deployment of an Efficient Golf Game Dynamics Management System Based on Flexible Wireless Technologies

**DOI:** 10.3390/s23010047

**Published:** 2022-12-21

**Authors:** Imanol Picallo, Erik Aguirre, Peio Lopez-Iturri, Javier Guembe, Eduardo Olariaga, Hicham Klaina, Jose Antonio Marcotegui, Francisco Falcone

**Affiliations:** 1Electrical, Electronic and Communication Engineering Department, Public University of Navarre, 31006 Pamplona, Spain; 2Tafco Metawireless, S.L., 31013 Ansoáin, Spain; 3Institute for Smart Cities, Public University of Navarre, 31006 Pamplona, Spain; 4Tecnologico de Monterrey, School of Engineering and Sciences, Monterrey 64849, Mexico

**Keywords:** golf, LoRaWAN, LPWAN technologies, radio communication, wireless sensor networks, ZigBee

## Abstract

The practice of sports has been steadily evolving, taking advantage of different technological tools to improve different aspects such as individual/collective training, support in match development or enhancement of audience experience. In this work, an in-house implemented monitoring system for golf training and competition is developed, composed of a set of distributed end devices, gateways and routers, connected to a web-based platform for data analysis, extraction and visualization. Extensive wireless channel analysis has been performed, by means of deterministic 3D radio channel estimations and radio frequency measurements, to provide coverage/capacity estimations for the specific use case of golf courses. The monitoring system has been fully designed considering communication as well as energy constraints, including wireless power transfer (WPT) capabilities in order to provide flexible node deployment. System validation has been performed in a real golf course, validating end-to-end connectivity and information handling to improve overall user experience.

## 1. Introduction

The trends in human settings have led during the past decade to the need of implementing more sustainable and efficient environments and the associated processes. This is especially true in the case of dense urban environments, such as cities, which by 2050 will account for 70% of the world’s population and demand up to 85% of all resources [1]. With this in mind, there has been an increasing effort in the implementation of Smart Cities, extended more recently to Smart Regions, in which resource usage is optimized, whilst increasing the quality of life of citizens, enhancing governance and administration at all levels (from municipal to supranational) and where new business models (e.g., exploitation of Open Data paradigms) are defined and proposed [2]. A smart city/smart region can be described as a system of systems, in which each one of these systems (e.g., Intelligent Transportation Systems-ITS, Smart Grid, Smart Health, Industry 4.0, water management, residue handling, etc.) takes advantage of information and communication technologies in order to provide context aware interactive environments. Moreover, each one of these systems, by means of interoperability principles can be coupled to a joint smart city platform, in order to further increase efficiency by taking advantage of the synergies among these systems (e.g., ITS interoperates with Smart Grid in order to plan electric vehicle charging, etc.).

Context aware environments require a high level of interactivity, considering that a large number of sensors and actuators can be present in a given scenario, as foreseen in Internet of Things (IoT) applications. In this sense, communication systems play a key role in order to guarantee information exchange, of a potentially very large number of users, with highly variable user requirements and constraints. Among the different types of communication systems, wireless communication systems play a key role, owing to their high deployment flexibility and inherent mobility. It is worth noting, however, that the operation of wireless links in these conditions is challenging, given increasing levels of interference by the variable system conditions in terms of bandwidth, bit rate/packet error rate, as well as by restricted energy availability (especially in massive wireless sensor network node deployments) and limited performance of embedded antenna systems (owing to form factor restrictions or the presence of elements such as the human body in wearables or the embedding elements, such as urban infrastructure or vehicles). With this in mind, multiple wireless communication standards have been developed in order to adapt to these highly dynamic conditions, whilst optimizing cost, size and energy usage. In the case of massive monitoring applications, wireless sensor networks have evolved from wide area network protocols (such as ZigBee) towards Low Power Wide Area Network (LPWAN) protocols (such as LoRa/LoRaWAN, Sigfox or MioTy and future IEEE 802.11ah protocols), as well to mobile network-oriented solutions (NB-IoT, LTE Cat. M1 and M2, 5G NR FR1 machine type communications, among others), with the idea of enabling massive telemetry/telecontrol connectivity in low to moderate transmission rate settings.

Sport activities, considering personal/collective training, competition monitoring or audience interaction, have been steadily taking advantage of the integration of sensors as well as information and communication technologies, being considered one of the systems within the Smart City/Smart Region framework. Multiple applications have been reported, such as monitoring biophysical/health parameters within sports practice [3], obtaining biophysical profiles of athletes [4], sensor fusion techniques applied in sports [5], novel sensor data extraction [6,7], analysis of real time kinematics [8], sensor data translation towards clinical analysis [9], sport classification aided by motion analysis and neural network support [10] or specific sport event tracking conditions owing to COVID-19 [11], to name a few.

In the case of golf, different approaches have been described in order to take advantage of information derived from sensors or the context in which the sport is developed. In [12], a review of dynamics models and measurement in golf is provided including technologies in order to measure the movements of the club, ball and players. In [13], a local sensor embedded in the grip end of a golf club is employed in order to analyze angular motion to aid golf player training. A golf ball with Radio-Frequency Identification (RFID) tracking capabilities is proposed in [14], in which photovoltaic cells are employed in order to enhance battery life and, hence, communication range of the system. Information related to golf swing segmentation is derived by inertial measurement unit (IMU) results and classified in different phases with the aid of machine learning techniques in [15]. Maintenance activities related to golf courses have also been described, by taking advantage of static wireless sensor networks combined with remote sensing data in order to monitor environmental parameters such as soil moisture [16]. Table 1 [13,14,15,16,17,18,19,20,21,22,23,24,25] summarizes some relevant contributions related to golf sport. It presents works focused on improving the movements of the golf player, specifically the swing motion. However, none of these manuscripts considers the golf course, its monitoring and the dynamics of the game as this work does, which allows an improvement of the playing experience throughout the golf course.

In this work, a monitoring system applied to golf sport development, related with the dynamic location of golf players within the course, will be described. The complete system, based on different types of nodes employing LPWAN connectivity, remote data processing and visualization capabilities, will be designed, implemented and tested in a real golf course. Wireless connectivity performance will be analyzed for the particular operational conditions of the golf course by means of in-house implemented 3D-Ray Launching algorithm (3D-RL) in order to provide volumetric assessment on wireless channel characteristics, combined with empirical/statistical analysis and wireless channel measurement campaigns. The complete device design will be described, considering connectivity requirements, location of players and energy handling, including wireless power transfer capabilities. Device as well as system level measurement results are presented, validating overall operational capabilities related with golf sport monitoring.

The work has been developed within the framework of the research project “Diseño y Desarrollo de Sistema de Comunicaciones para la Gestión eficiente de campos de golf (T-Golf)”, funded by the Government of Navarre, developed by Public University of Navarre and TAFCO Metawireless [26], with the collaboration of Castillo de Gorraiz Golf Club. Following the needs detected by the club, the main objective of the project is to design an optimal and flexible communications system to efficiently manage the dynamics of the game and provide precise information to users and managers at any golf course, and specifically:To design and develop a wireless sensor network through a non-commercial pilot installation, capable of speeding up and evaluating playing time, informing the user in real time and monitoring the status of golf courses.To assess and select the communication protocols and generically define the network topology that best suits the needs of the project, in terms of flexibility, overall performance and cost.To define and design a flexible hardware, adaptable to the needs of each field, developing the prototypes of devices, these being: light, low consumption, robust, and at the same time competitive in cost.To design and develop software (application level) capable of working in real time with the information obtained from the sensor network, offering the players real-time and historical data of the round, game manager tools to help solve slow game, and maintenance staff information of the state of the course in order to optimize resources.

The paper is organized as follows: Section 2 presents the real golf course where the study and deployment have been done. Section 3 is devoted to radio propagation assessment of the golf course, where a single hole has been analyzed first and then the entire golf course has been assessed by means of measurements employing two different wireless technologies: ZigBee and LoRaWAN. In Section 4, the developed prototype is presented, in terms of hardware and software implementation. Finally, Section 5 discusses the conclusions.

## 2. Description of the Golf Course

The club where the study has been carried out is the Castillo de Gorraiz Golf Club, located in Gorraiz, Navarre, Spain. The golf course has an approximate area of 450,000 m^2^, with altitude levels ranging between 477 m and 517 m, and paths with a maximum vertical area of approximately 1350 m and a horizontal area of 1140 m. Figure 1 shows a view of the golf course under analysis. As it is highlighted, there is a high elevation area which is expected to affect significantly the wireless links. Moreover, the Club House is also on an elevation (see starting point in Figure 2b,c, which in principle could be beneficial for the coverage if a network coordinator or gateway is deployed there, due to Line of Sight (LoS) or partial LoS conditions. The scenario under study is made up of undulating terrain (soft hilly terrain), with the presence of vegetation of moderate density and height. In order to gain insight on the orography of the course, Figure 2 presents four different elevation profiles starting from the Club House. The environment (including the orography and the tree mass and foliage) and holes distribution are expected to be key elements to consider for the deployment of the wireless networks, especially for Holes 3, 4, 5, 6 and 14, since the expected operating frequencies of the wireless communications systems (range between 430 MHz and 2400 MHz) represent an additional loss term due to the dispersion effect of the radioelectric signal.

## 3. RF Propagation within Golf Environments

With the purpose of studying both the topology of the wireless network and its optimized configuration, the radioelectric planning tasks in the scenario under analysis are presented in this section. First, the study has been focused on a single hole. The objective of this study is to analyze the radio propagation behavior within this kind of environments for a better understanding of its characteristics to help the development of potential future applications, such as providing the exact localization of the hole/flag within the green directly and wirelessly to the golf players’ hardware devices (changing the holes’ locations within the green every 15 days is a common practice), via short/medium range wireless communication technologies. The presented measurements validate the employed simulation methodology, making this approach appropriate to be used for the analysis of every hole (which are different and unique), not only of the presented golf course, but also for every golf course’s holes around the world, without the need of taking measurements or being present in them.

Then, a broader and more complex study for the entire golf course is presented, where a wide area network has been tested, since the characterization of the radioelectric channel and the subsequent estimation of coverage/capacity ratios depend strongly on the type of environment, being site-specific.

### 3.1. RF Assessment of a Single Golf Hole

The assessment of the radio propagation within a single hole has been carried out for Hole number 1 (see Figure 3a). Figure 3b shows the Google Earth picture of Hole 1. Its area is approximately 7200 m^2^, and its elevation profile is shown in Figure 3c, where the maximum height difference is around 8 m, being the green the highest zone.

Measurements as well as deterministic simulations by a broadly used and validated 3D Ray Launching software developed in-house have been performed [27,28,29]. Frequency bands of 900 MHz and 2.4 MHz have been analyzed, in order to cover the most common wireless communication technologies employed for LPWANs and WSNs (LoRaWAN Europe and USA, ZigBee).

But first, the existence of interfering electromagnetic noise or external radio signals has been studied. For that, a portable spectrum analyzer has been employed in field, and the obtained results are presented in Figure 4. Significant power levels have been detected for both frequency bands. Thus, further analysis and validation of the proposed system will be mandatory in order to ensure an adequate performance of the network deployed on the golf course.

The RF measurements have been carried out throughout Hole 1 for the 900 MHz and 2.4 GHz frequency bands in a linear path from the transmitter to the hole, as shown in Figure 3c, covering a distance of 166 m. A Voltage Controller Oscillator (VCO) was used as a transmitter at the frequencies of interest (models ZX95-1700W and ZX95-2500 from Mini-Circuits^®^, Brooklyn, NY, USA), which provide 9 dBm and 7.5 dBm of transmission power, respectively. The antenna OmniLOG 30,800 has been employed, which is a compact omnidirectional antenna with a variable gain depending on the frequency operation in a wide range from 300 MHz to 8 GHz. The Agilent’s FieldFox N9912A portable spectrum analyzer was used to measure the received power level at the height of 1.1 m from the grass. In Figure 5, pictures of the measurement setup for Hole 1 are shown.

For the deterministic simulations, a scenario as close to Hole 1 as possible has been created by the 3D-RL software (see Figure 6). The orography and the electrical properties (permittivity and conductivity) of the materials (mainly grass and wet grass) have been considered for the simulation. Table 2 shows the configured simulation parameters, based on the RF measurements and previous algorithm convergence studies [30,31]. It is worth noting that for this specific case (a very large and sloppy scenarios with almost no obstacles), a specific analysis has been performed to adjust the angular resolution of launched rays in order to obtain more accurate simulation results, finally setting the parameter to 0.5°.

Thanks to one of the main features of the 3D-RL tool of obtaining 3-dimension results, any point of the whole volume of the scenario can be analyzed with a single simulation while the measurements provide results only for the specific measurement points. This is especially useful in scenarios like this golf course, with significant height differences. Figure 7 shows the estimated distribution of RF power depicted in 2D cut planes for different heights. Simulation results for the frequencies of 975 MHz (Figure 7b) and 2.4 GHz (Figure 7c) are presented, corresponding to the planes at the heights marked in Figure 7a. The planes at height 3.5 m and 8.5 m correspond to the height of the transmitter and the flag, respectively. It is worth noting how the RF power distribution presents rapid variations typically due to multipath propagation, in this case due mainly to the ground effect and the unevenness of the terrain. As a detail, it can be observed how the RF power decreases drastically when the signal goes underground (the dark blue zone of planes at height 3.5, 4.5, 5.5 and 6.5 m). The shorter range for 2.4 GHz can be also observed, due to the higher propagation losses compared with 900 MHz band.

Once simulation and measurement results have been obtained, a comparison between them is presented in Figure 8, with the aim of validating the simulation estimations. For the comparison, the different gains of the employed OmniLOG 30,800 antenna have been applied to both frequency bands. This is the reason why the 900 MHz band presents lower RF power levels (i.e., the antenna performs better at 2.4 GHz). As expected, both graphs show how the power level decreases with the distance. From 80 m on, which is when the slope begins to go upwards (see Figure 3c), the curve stops following that tendency, since the receiver antenna starts being more affected more with multipath components due to the hill. It is important to note that the divergence appreciated from 110 m is due to the distance and the cuboids size employed in the simulations.

Regarding the comparison, the obtained error mean for 900 MHz band is 2.75 dB and for 2.4 GHz, 2.88 dB, showing good agreement between measurements and simulation results. Thus, validating the simulation methodology for radio planning tasks in this specific environment.

### 3.2. RF Assessment of the Entire Golf Course

After analyzing the radio propagation characteristics for different operating frequencies within a single golf hole of a real golf course, which provides interesting information for a future deployment of wireless communication systems and applications, a broader study has been performed in order to assess the RF propagation behavior for an entire golf course. For that purpose, two well-known and well-established wireless communication technologies have been employed: ZigBee and LoRaWAN. This study aims to assess these two different technologies for their use in real golf course scenarios, as well as to evaluate them for their implementation in the solution developed in our project, and presented in this paper.

Both ZigBee and LoRaWAN are wireless technologies broadly used for the deployment of WSN and monitoring applications [32,33,34,35]. The main differences are that LoRaWAN could reach longer distances due mainly to its better sensitivity level. The main characteristics of ZigBee and LoRaWAN are summarized in Table 3.

For the study, a network with multiple devices has been built, for both communication technologies. Figure 9 shows schematically the network topology employed in this study for LoRaWAN (Figure 9a) and ZigBee (Figure 9b). The devices, or nodes, send the geographical location (obtained by GPS) to a central node, where this information is processed and displayed on a PC or laptop in order to be monitored. Due to their mobile nature, the nodes are battery powered.

The prepared LoRaWAN network operates at 868 MHz, and consists on nodes based on different commercial hardware modules. Specifically, each node consists of a STM32-NUCLEO-L073Z (STMicroelectronics N.V., Geneva, Switzerland) board with the microprocessor, a geolocalization board X-NUCLEO-GNSS1A1 (STMicroelectronics N.V., Geneva, Switzerland) with GPS antenna, and a communication shield LORA I-NUCLEO-LRWAN1, all the elements from ST Microelectronics.

On the other hand, the ZigBee network operates at 2.4 GHz, and the nodes consist on the same microprocessor and geolocalization boards used for the LoRaWAN nodes, and an I/O Expansion Shield V7.1 (from DFRobot) with a Digi’s XBee 3 Pro mote for ZigBee communication. Figure 10 shows the implemented nodes for LoRaWAN (right) and ZigBee (left).

Since all the nodes are battery powered, a charging control system will be required. For that purpose, the LTC4040 circuit has been implemented, which allows the management of different types of batteries. A prototyping board has been designed and developed by TAFCO Metawireless that allows the placement of a node and a battery managed by the aforementioned circuit. Figure 11 presents the board and a node mounted on it, as an example. The battery is a 2600 mAh Li-Ion battery. The circuit provides to the node a constant voltage of 5 V.

Regarding the gateways of each network, a node like the others has been employed for the ZigBee network, configured as a network Coordinator connected via USB cable to a laptop. For the LoRaWAN network, a commercial Gateway has been employed: Laird RG186. This model has been selected due mainly to two reasons: its connectivity flexibility (WiFi, Ethernet and Bluetooth) and its ease of configuration, a valuable characteristic since TAFCO Metawireless’s objective is to create a solution including a comprehensive platform capable of providing all the required functionalities for the specific application developed in the T-Golf project. Additionally, it is important to note that the Laird manufacturer has in the catalog equivalent ruggedized models, which comply with the IP67 standard, for the final outdoors deployment of the system.

Finally, a server has been implemented in TAFCO facilities. It consists of a NODE JS server that uses a MongoDB database. The server stores the packets sent by the gateways, and the collected data is shown in a map. For that, several HTML pages have been configured, where the location of the nodes are easily monitored, alongside the received RF power level, the coordinates of the node, the packet arrival time, the packet number, the nodes ID and other parameters such as the Spreading Factor (only for LoRaWAN) or the transmitting power level.

#### Deployment on the Golf Course

Once the nodes and gateways for both ZigBee and LoRaWAN networks are configured, a measurement campaign has been carried out in the Castillo de Gorraiz Golf Club presented in previous sections. The aim of this first test on the course is to check and validate the designed configuration for the solution, as well as to compare both wireless technology alternatives.

The gateways have been deployed in front of the Club House, as can be seen in Figure 12. It is important to note that for the LoRaWAN Gateway, a specific antenna has been included. This antenna, an OMB.868.B08F21 from Taoglas is an outdoor high gain (8 dBi) antenna, designed to operate at 868 MHz. The antenna for the ZigBee gateway is a 2.1 dBi gain antenna, model A24-HASM-450, designed to operate at 2.4 GHz band.

On the contrary, the nodes, which are mobile devices, have been deployed along all the 18 holes of the golf course, following a hypothetical path traveled by a golf player.

The results for the LoRaWAN network are presented in Figure 13a. The colored points represent the location of the mobile node and the RSSI level received at the gateway. The area of Holes 3, 4 and 5 (right side of the map) is the one with the lowest RSSI due to the fact that they are behind a high elevation zone (the big mass of trees), which causes the signal level to drop. Although the wireless communications in that zone can be critical, and connectivity could be lost between the node and the GW if the conditions worsen, given the sensitivity of the GW LoRaWAN (−148 dBm), the transmitted packets reached the GW. In addition, Figure 13b,c show the RSSI and SNR of a selection of points in order to display some specific values for different of the golf course.

Regarding the deployment of the ZigBee-based network, Figure 14 shows the obtained results. Opposed to LoRaWAN, a single ZigBee Gateway (i.e., the coordinator) is not able to cover the entire golf course. As can be seen in the figure, the coordinator and the deployment of two extra ZigBee routers (acting as repeaters or coverage extenders) have been needed to ensure the wireless connectivity for the whole area of the course: a first approximation was made, deploying a single router, but as can be seen in Figure 14, and due to the high elevation area (shown in Figure 1), a router could not cover the area behind that elevation area and at the same time have connectivity with the coordinator. The colored polygons represent approximately the area covered by each ZigBee Router. The yellow one corresponds to the gateway, and the blue and the red, to router 1 and 2 respectively. In order to gain insight on this matter, Table 4 presents the packets received by the ZigBee coordinator and the two ZigBee Routers during the measurement campaign, which consists of an emulation of a round of golf. The packets have been sent every 5 s, and the round lasted 3 h and 30 min approximately. All the sent packets were received by the coordinator (due to the successful network deployment and the retransmissions provided by the ZigBee standard) and could be seen in the cloud. Specifically, a total of 2386 packets were sent and received, where 2091 were received directly by the coordinator, 215 through Router 1, and 80 through Router 2. Note that both routers received more than those packets, but duplicated packets (e.g., received by the coordinator and Router 1) are not considered.

Based on the presented results, both wireless technologies are valid to cover the entire golf course, but the following two main reasons lead us to select LoRaWAN for our final solution:There is not any infrastructure on the course to supply energy to static nodes, so providing energy to ZigBee routers will require an extra installation of an energy source such as solar panels, while LoRaWAN could be deployed without the need of repeaters.The energy consumption of ZigBee nodes is much bigger than LoRaWAN devices, and considering that the nodes’ batteries should last at least a day (at night will be recharged), this is a major issue for our solution.

Finally, as mentioned, the presence of the high elevation zone could lead to the loss of connectivity during a round of golf. Therefore, a final, more robust solution could require the deployment of an extra LoRaWAN gateway, as will be seen in the next section.

## 4. Prototype Design

Once the radio planning analysis of the golf environment has been carried out, this section details the design of the prototype from the hardware (HW) and software (SW) point of view. In addition, the final prototype and tests during a round of golf on the course are presented, validating the implemented whole system.

### 4.1. Hardware Design

The following points summarized the elements desired from the HW side:Reflective thin-film transistor (TFT) display, with low consumption technology.Global Navigation Satellite System (GNSS) navigation system.LoRaWAN communication.Smart battery charge/discharge.Button management.Wireless charging.

The designed and manufactured prototype has been called USER-ED, which is the portable device the golf player will be carrying during a golf round. It consists of a visualization display, where the player can count the number of strokes made in each hole, among other capabilities. The device is provided with all the maps of the greens of the golf course. Through the GNSS geolocation module included in the device, the player can see the distance from his position to the green of the current hole. On the other hand, the player can also observe the time of the game and the distance that has been covered. The device automatically detects the player’s presence near the green, showing him the score entry screen automatically.

The most important HW elements of the USER-ED devices are explained below:3.For the navigation system, the L86-M33 GNSS module from Quectel has been implemented. It incorporates an embedded antenna and low-noise amplifier (LNA). It combines many advanced features that are beneficial to accelerating Time to First Fix (TTFF) improving sensitivity and consumption. It also supports various positioning, navigation and industrial applications including autonomous GPS, GLONASS, SBAS, QZSS and AGPS.4.Concerning Lora communication, a CMWX1ZZABZ module from Murata is implemented. It supports LoRaWAN long-range wireless protocol and incorporates a Semtech SX1276 transceiver and a STMicro STM32L0 series ARM Cortex-M0+ 32-bit microcontroller (MCU). A 2.63 dBi chip-type antenna from ABRACON has been selected for this device. It is a surface-mount technology (SMD) component, so it is easy to incorporate into the manufacturing line. In addition, it improves the usability of the device avoiding the use of an external antenna.5.The chosen microprocessor is an ultra-low power ARM Cortex M4 of 32 bits, with 1 MB of flash memory and 128 KB of SRAM from the manufacturer ST. Among many other features, it presents multiple communication interfaces that are very useful for our application, such as USART, SPI and I2C ports, to name a few.6.Regarding the WPT module, the USER-ED has been designed in order to contain a medium-power WPT system, required at the first stages of the project by the Castillo de Gorraiz Golf Club managers. In this work, the Semtech TSDMTX-19V2-EVM has been chosen for wireless charging. The Semtech TSDMTX-19V2-EVM is a wireless charge transmitter based on the Semtech TS80000 Wireless Power Transmitter Controller, TS61002 FET Controller, TS30011 DC/DC Converter, TS94033 Current Sense Amplifier and SC32300B Controller. This evaluation module is a demo platform for testing up to 15 watts of wireless power transfer that compliance the dominant Qi and Power Matters Alliance (PMA) standards.

The WPT smart management is carried out by programming the TS80000. First, this controller searches for a receiver, and when it is found, the receiver informs the transmitter of its power requirements and the power transfer begins. Second, the system verifies that the correct amount of power is being sent and that none is being lost due to Foreign Object Detection (FOD) system. The receiver continuously provides power requests to maintain the energy transference. If the request ends, the transference ends as well. The transmitter can provide varying amounts of power at different times, as requested by the receiver through this protocol. If the receiver does not need more power, such as when the battery is fully charged, then the transmitter reduces its output accordingly. From the point of view of the receiver module, it has to consist of a wireless charging receiver that includes the management of a battery with the following characteristics:LiPO battery of at least 2000 mAh (required).Management of the battery charge: constant output of 5 V and up to 2 Amps when not charging (necessary).Inclusion of some kind of battery charge indicator accessible by I2C or similar (desirable).USB charging option (desirable).

Among different options, the Semtech TSDMRX-5V/10W-EVM (Newbury Park, CA, USA) receiver module has been selected to be implemented in the USER-ED devices, since it meets all the requirements and it is compatible with the TSDMTX-19V2-EVM transmitter.

The empirical tests on laboratory showed that the WPT systems worked correctly and without suffering cuts. Then, as the USER-ED devices must be encapsulated, different materials for the encapsulation have been tested. Table 5 presents the analyzed materials, and showed whether the WPT system worked for the required distance between the transmitter and the receiver coil. The ‘Ok’ corresponds to a battery charge without interruptions. As can be seen, several materials comply with the required distance, but due to cost and ease of manufacturing (in this case, by a 3D printer), PLA has been chosen for the prototype encapsulation. Figure 15 presents the prototype.

### 4.2. Software

Regarding software programming, two different applications have been implemented. The first software application is responsible for the management of the LoRaWAN communications (receiving packets, storing them, etc.). This app is based on a database which resided on the server. The second is a user interface software application, where the different management activities are collected, as well as game events.

For the development and design of these software applications, the platform known as the MEAN stack (acronym for: MongoDB 6.0, ExpressJS 4.18.1, AngularJS 1.8.3, NodeJS 18) is used. As a summary, a diagram of the software tools used for the client and server part is shown in Figure 16, which is an improvement over the simpler platform developed for the first measurements presented in Section 3.2.

The user interface software is the main app developed in this work, where four different user profiles have been defined: system administrator, course administrator, maintenance technician and player. Each of them contains different menus on the same platform in order to guarantee access only to certain system resources and information. Figure 17 shows some examples of management screenshots of the developed software for the different user profiles.

On the other hand, the software of the USER-ED provides the golf player with the required information during the golf round: the entire golf course map, all the holes map individually, the localization of the player, the distance between the player and the green, and the PAR of the hole and the possibility of introducing manually the number of strokes (only available when the player enters the green of the corresponding hole), just to name the most relevant ones. Figure 18 shows a picture of a USER-ED showing the map a hole and two screenshots for different functionalities.

### 4.3. System Validation

Once all the prototype nodes and software have been checked and tested in laboratory conditions, the real deployment of the pilot has been carried out in the Castillo de Gorraiz Golf Club. In order to test the system under real conditions, four golf rounds have been emulated, using four different USER-ED devices, i.e., four golf players have been present on the course at the same time. It is worth noting that the presented solution is limited, in terms of number of users, by the chosen wireless technology. In this case, where LoRaWAN has been employed, the maximum nodes operating in the same network (including USER-ED and potential sensors deployed on the golf course for other applications such as monitoring of the state of the greens) will be given by the messages supported by the gateway over the course of a 24-hour period. Usually, if each node sends, for example, 100 messages a day, such a gateway can support about 1000 devices. In case ZigBee technology was chosen, the maximum number of devices/nodes connected to that wireless network will be 65,535. Figure 19 presents several pictures taken during the test under real conditions. Since we saw that the coverage with a single LoRaWAN gateway could be problematic for some holes (see Section 3.2), and extra gateway has been deployed in order to reinforce the coverage in those areas. As can be seen in Figure 20, the coverage for each gateway alone produces packet losses in some areas of the course (Figure 20a,b), while the deployment of the two gateways provide a very robust coverage for the entire golf course (Figure 20c), not losing connectivity even when the four devices are operating at the same time (Figure 20d).

In the same way, the software, run on a laptop, presented the golf rounds’ information (stored in the server) as well as the locations of the players in real time without problems (Figure 21). The only incident during the emulation of the golf rounds was a USER-ED which turned off due to problems with the smart battery charger circuitry.

## 5. Conclusions

In this work, a golf game dynamics management system based on wireless communication technologies has been designed, assessed and deployed in the Castillo de Gorraiz Golf Club. The implemented ad hoc wireless network enables the course manager to know the times of all the players in each one of the holes on the golf course. Likewise, the devices implemented in the system offer information to the player on the distances to the green and the counting of the number of strokes for the follow-up of the game.

The developed hardware devices (USER-ED) have been designed and oriented for their application in golf courses, with adequate dimensions for handling by the player both at the level of manual use and to be installed on a club cart or a buggy. The device implements a display that offers excellent viewing quality in outdoor environments and sufficient information for the development of the game. The consumption of the device is optimized to comfortably cover the duration of a game, benefiting from the low consumption of the display.

In the same way, a software has been designed that satisfactorily meets the needs of both the players (real-time position with respect to the green, counting strokes for each hole, elapsed playing time, distance covered, etc.), as well as the game manager or course manager (analysis of slow game with indication of times of all players in each hole).

Additionally, the conducted exhaustive study in terms of radio planning by means of both simulations and measurements has led to gain insight on the deployment methodology for golf courses, which will be site-specific due to the inherent characteristics of each golf course (orography of the terrain, presence of vegetation, etc.).

The pilot installation on the real golf course was carried out with the developed application running in the cloud, allowing its interaction from any device (desktop computer, laptop, mobile phone or tablet) connected to the Internet. The successful deployment of the whole system and the employed methodology allows replicating them in order to be installed in any other golf course, adapting the system to the specific characteristics of each course. Finally, it is important to highlight that the proposed system is very flexible in terms of adding new functionalities such as monitoring of environmental variables (temperature, humidity, atmospheric pressure, anemometer, rain gauge), soil parameters (temperature and humidity, conductivity, salinity and dielectric constant) or implementing management tools associated with maintenance personnel (position of machines, notices to personnel, etc.).

## Figures and Tables

**Figure 1 sensors-23-00047-f001:**
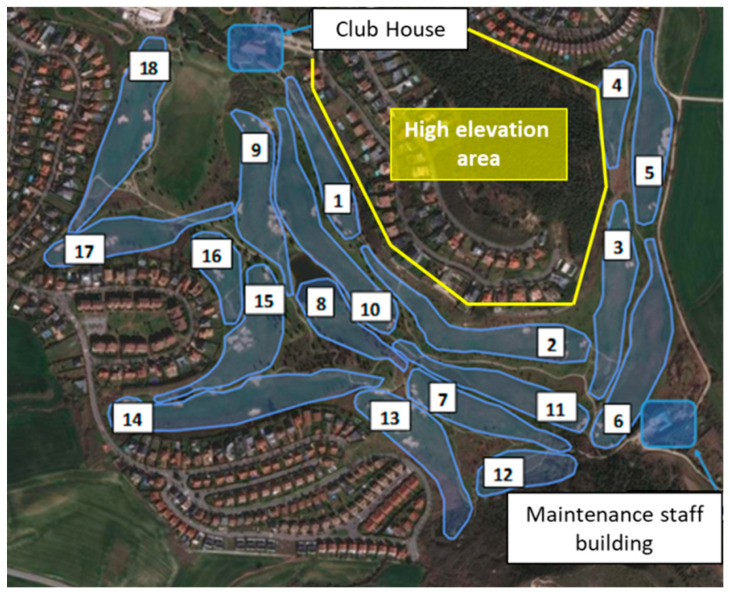
The golf course with 18 holes under analysis.

**Figure 2 sensors-23-00047-f002:**
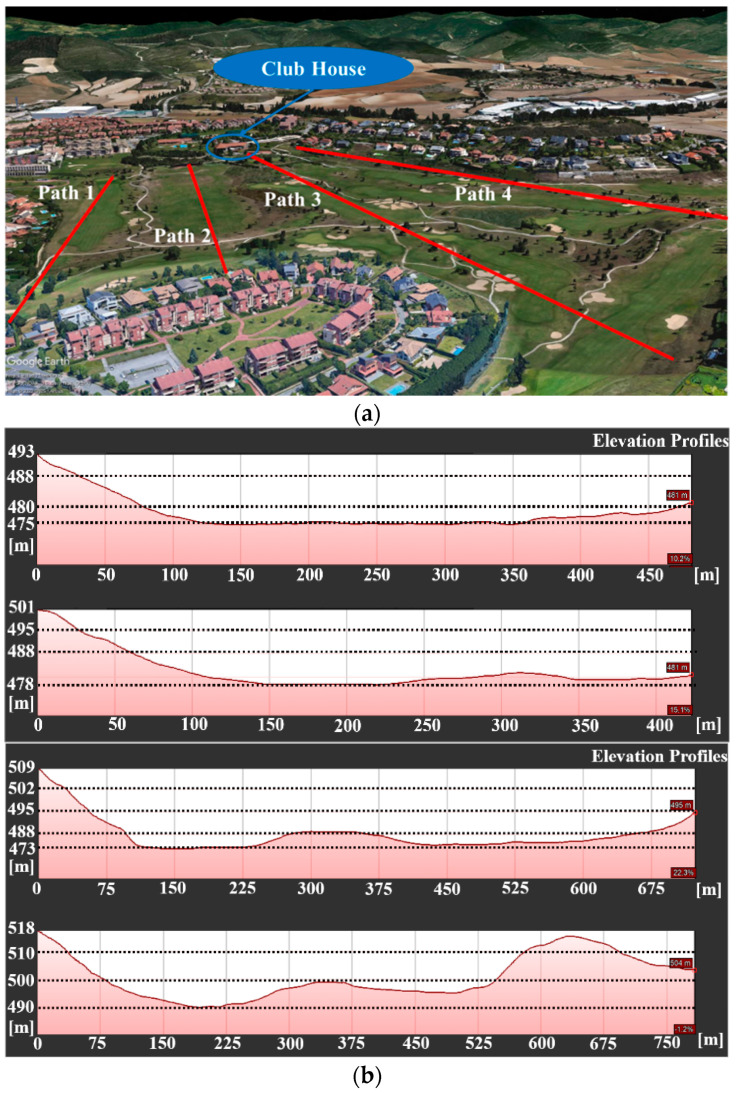
Elevation profiles of the golf course: (**a**) General Google Earth picture of the course; (**b**) Elevation profiles for Paths 1, 2, 3 and 4.

**Figure 3 sensors-23-00047-f003:**
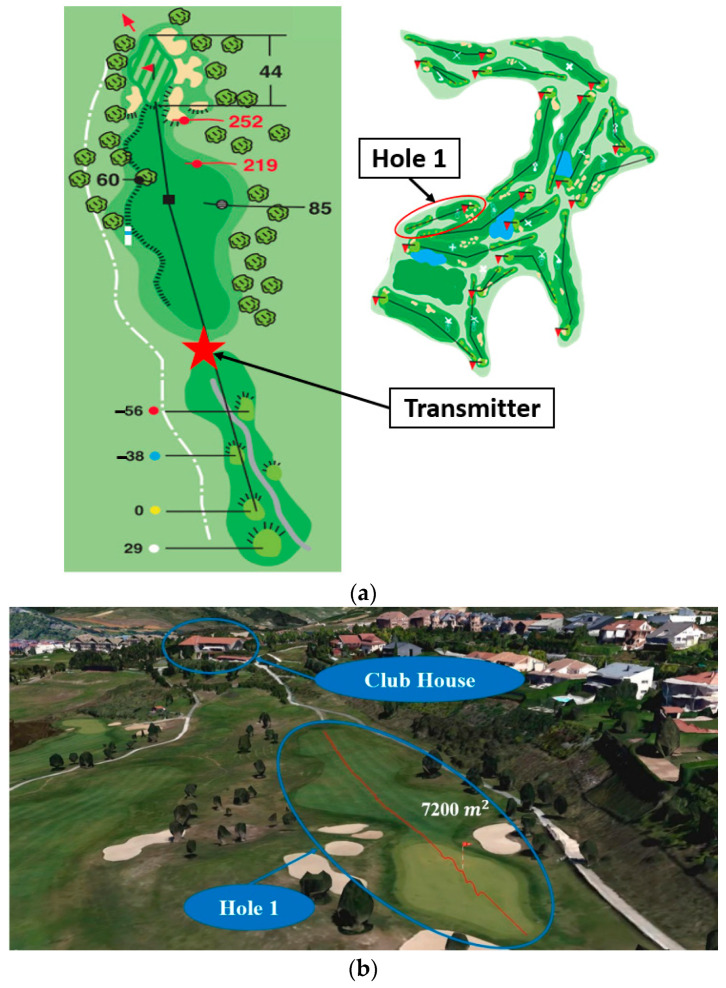
(**a**) Plane of Hole number 1 of the Golf Course; (**b**) Google Earth picture of the extension area of Hole 1; (**c**) Elevation profile of Hole 1.

**Figure 4 sensors-23-00047-f004:**
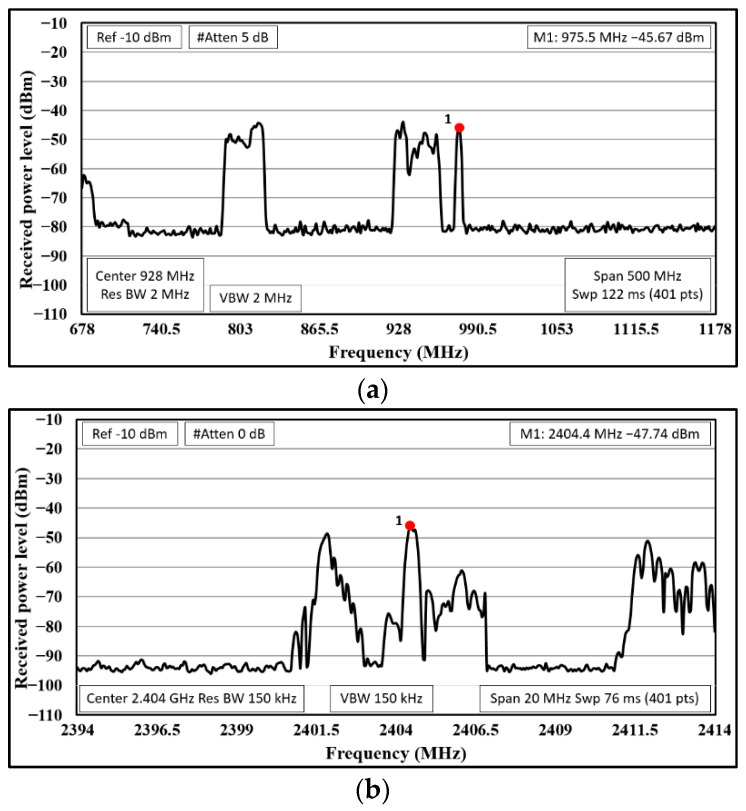
Existing signals in the frequency bands of interest within Hole number 1: (**a**) around 900 MHz; (**b**) around 2.4 GHz.

**Figure 5 sensors-23-00047-f005:**
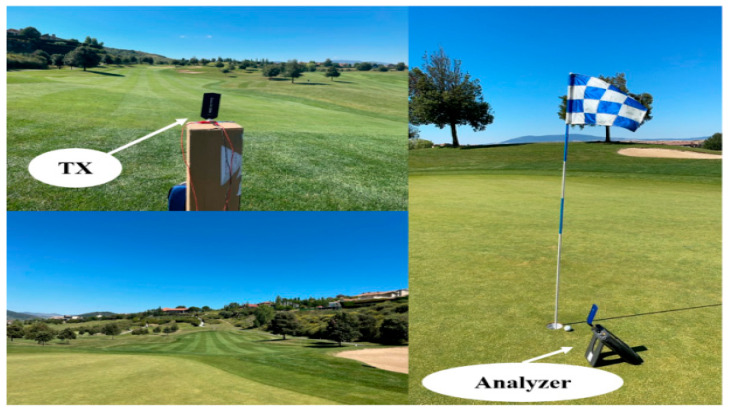
View of Hole 1 and the measurements setup.

**Figure 6 sensors-23-00047-f006:**
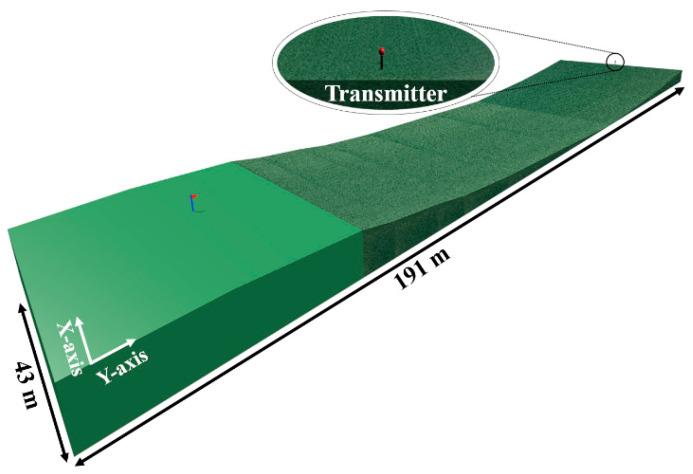
Hole 1 scenario created using the 3D-RL tool.

**Figure 7 sensors-23-00047-f007:**
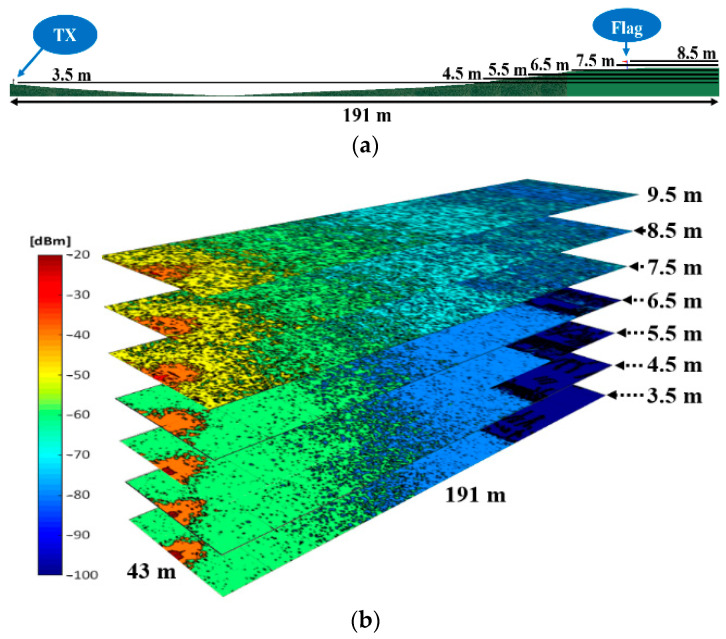
Estimated 2D planes of the received power level at different heights: (**a**) Lateral view of the created scenario; (**b**) Cut planes (XY) in the 975 MHz band; (**c**) Cut planes (XY) in the 2.4 GHz band.

**Figure 8 sensors-23-00047-f008:**
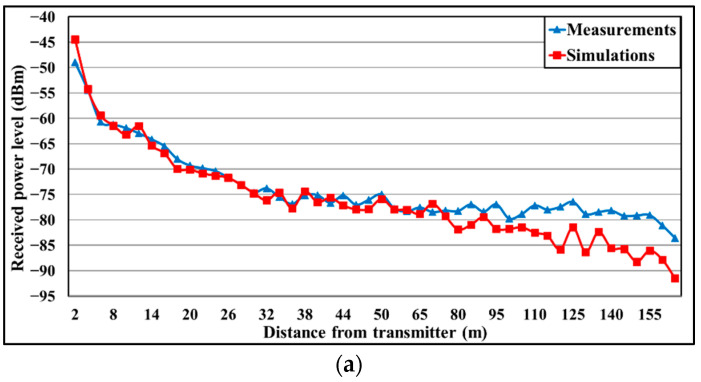
Comparison between RF measurements and simulations using the 3D-RL tool for the frequency bands of: (**a**) 975 MHz; (**b**) 2.4 GHz.

**Figure 9 sensors-23-00047-f009:**
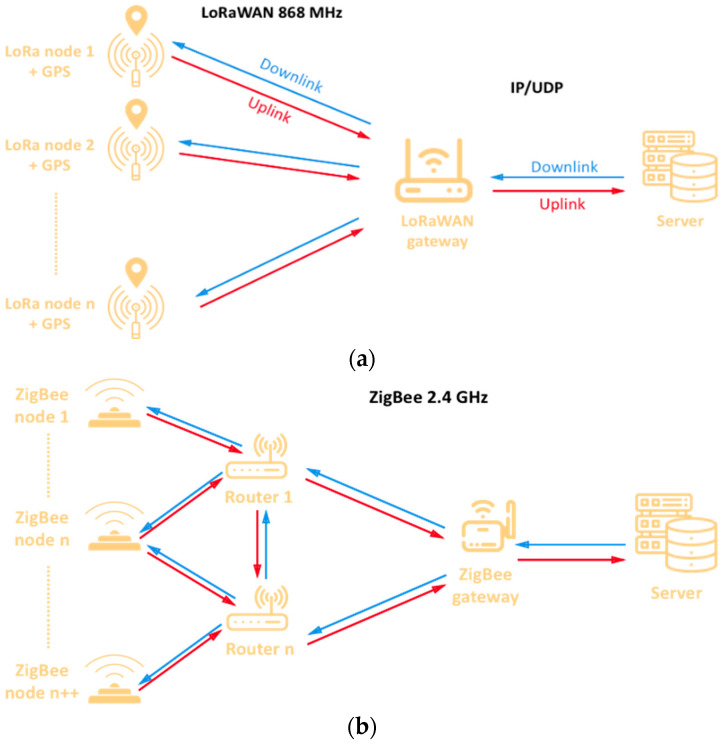
Schematic representation of the network topologies to be deployed on the golf course, (**a**) LoRaWAN and (**b**) ZigBee.

**Figure 10 sensors-23-00047-f010:**
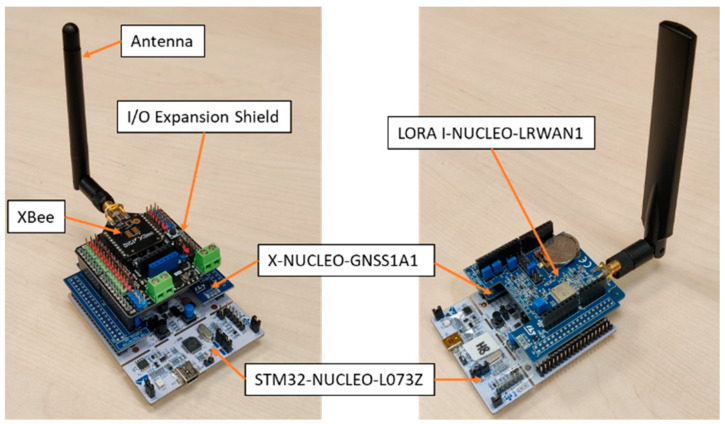
Implemented nodes: ZigBee (**left**) and LoRaWAN (**right**).

**Figure 11 sensors-23-00047-f011:**
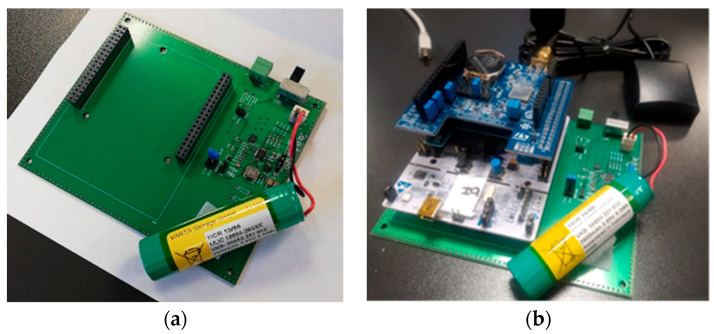
(**a**) Prototyping board and (**b**) a node mounted on the board ready for measurements.

**Figure 12 sensors-23-00047-f012:**
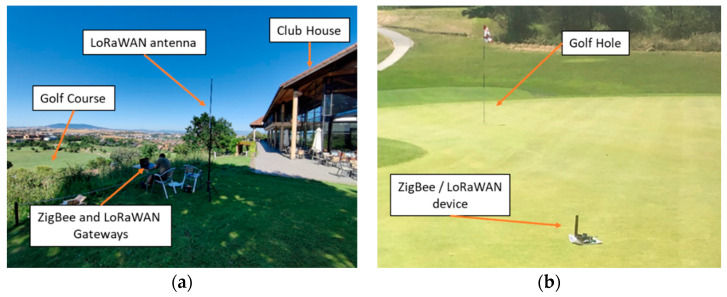
(**a**) Gateways deployment in front of the Club House; (**b**) Node deployment on the Green.

**Figure 13 sensors-23-00047-f013:**
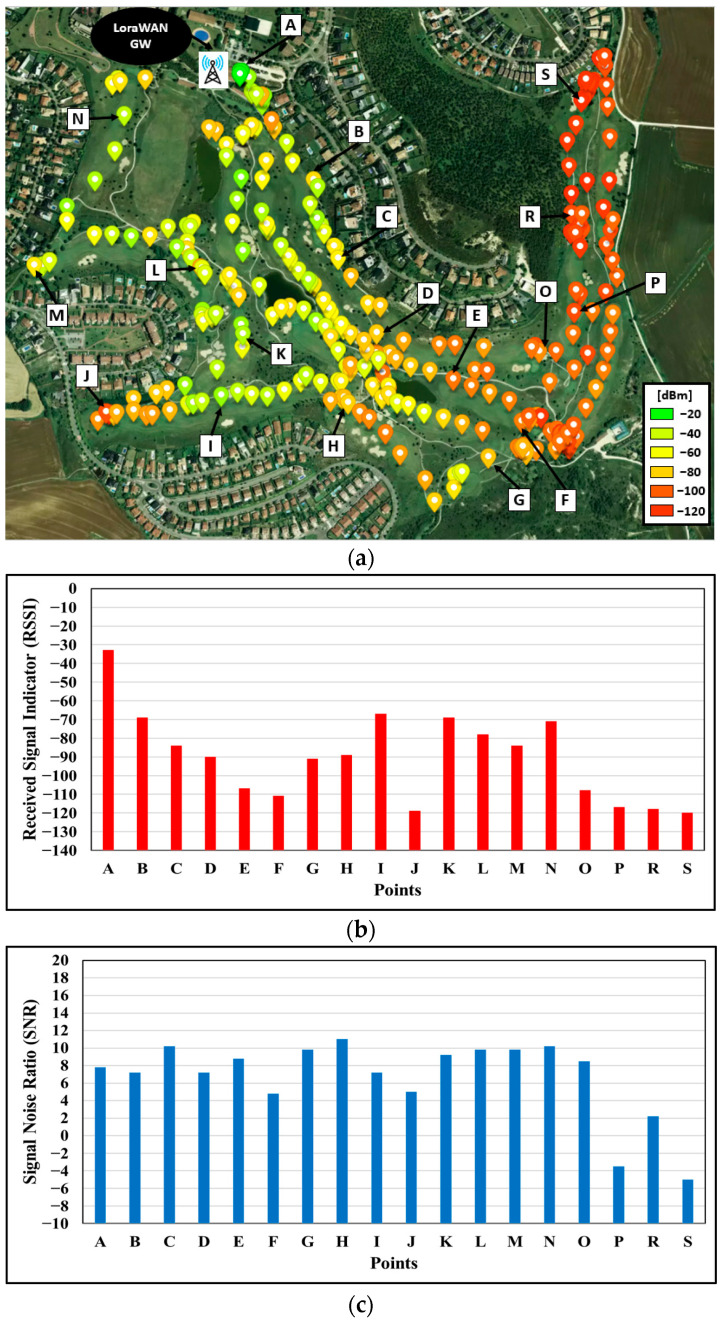
(**a**) Location of devices when packets are received by the LoRa WAN Gateway, with the corresponding RSSI level; (**b**) Detail of the RSSI level for the points corresponding to the points labeled in Figure 13a; (**c**) Detail of the SNR for the points corresponding to the points labeled in Figure 13a.

**Figure 14 sensors-23-00047-f014:**
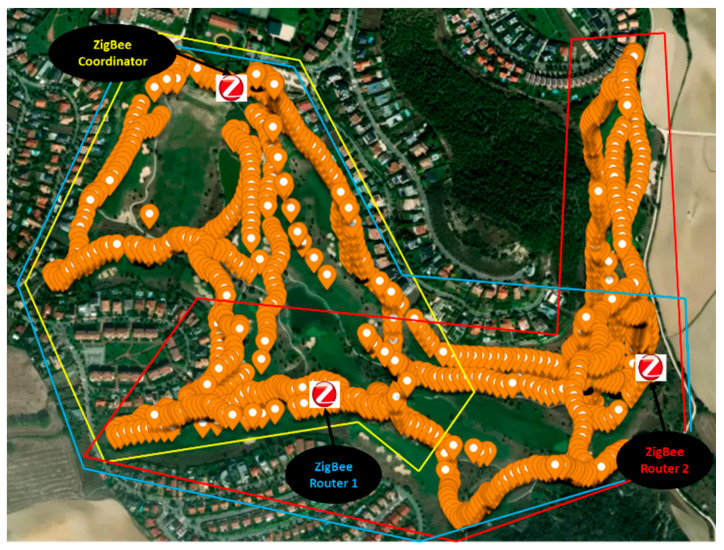
ZigBee coverage for the golf course.

**Figure 15 sensors-23-00047-f015:**
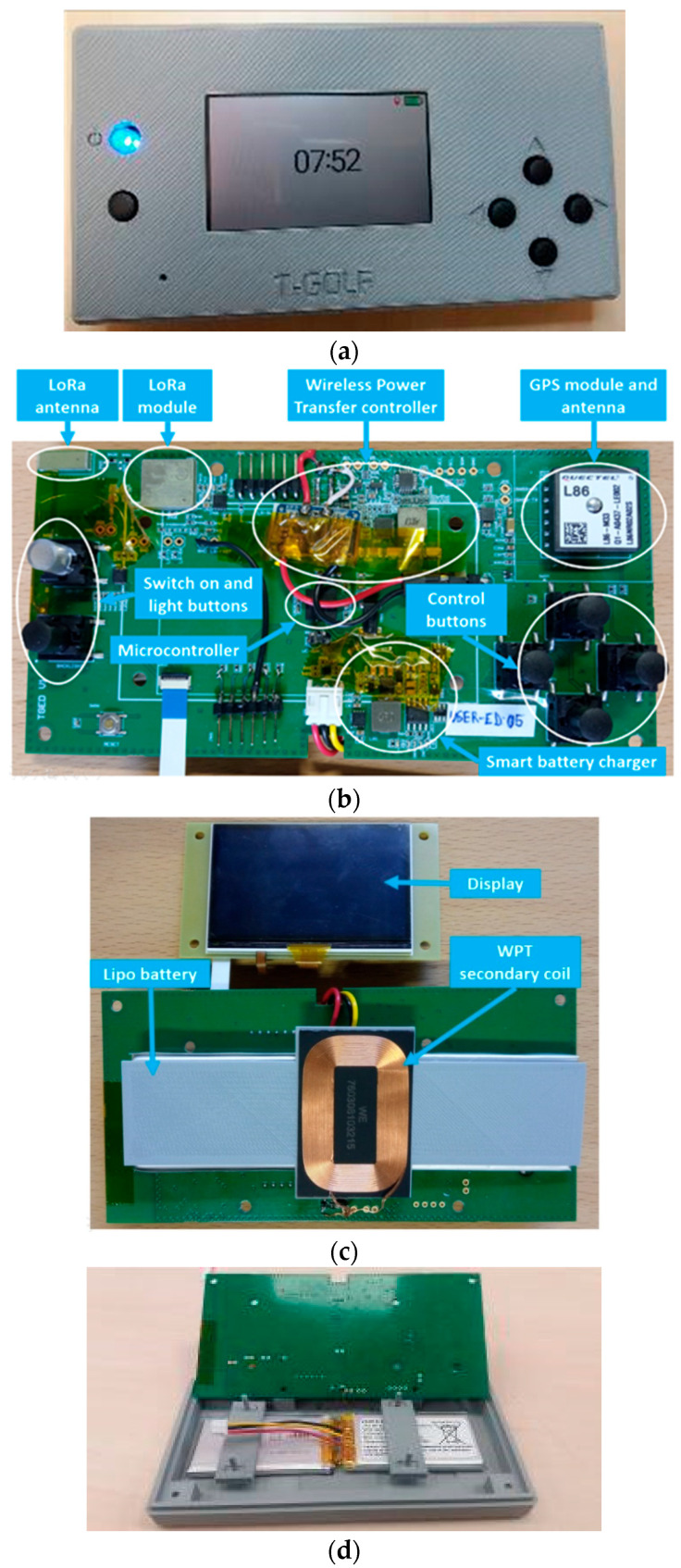
USER-ED prototype: (**a**) view of the encased prototype; (**b**) top view of the electronic board; (**c**) bottom view of the electronic board; (**d**) batteries and encapsulation.

**Figure 16 sensors-23-00047-f016:**
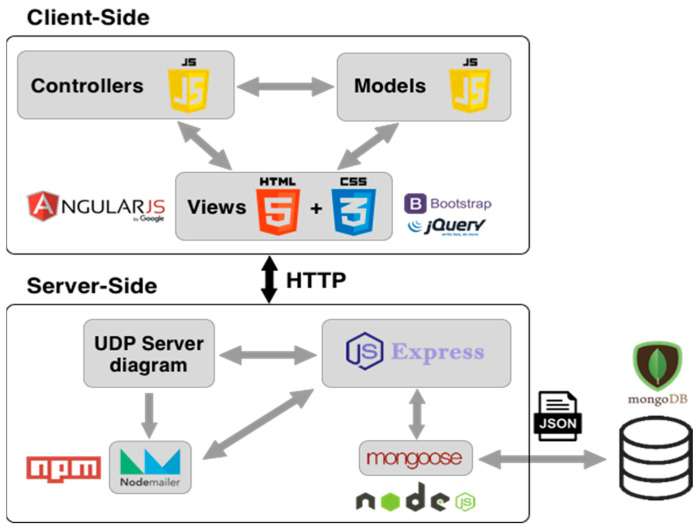
Diagram of the platforms employed in the implemented software solutions.

**Figure 17 sensors-23-00047-f017:**
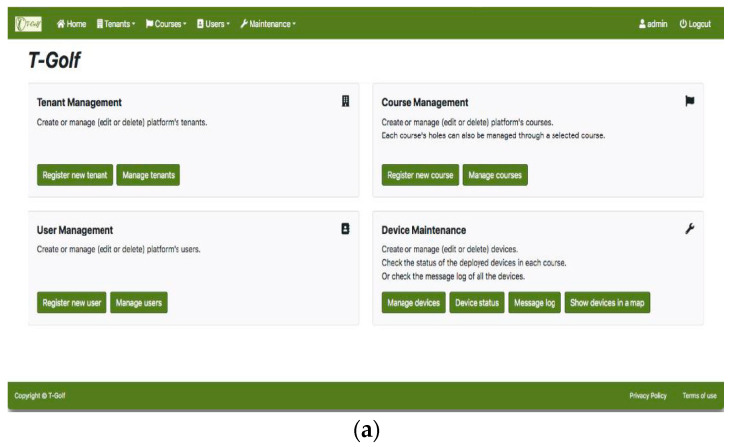
Screenshots of the interface for different user profiles and applications: (**a**) Management screen for the System Administrator profile; (**b**) All players’ location in real time for the Course Administrator user profile; (**c**) State of the golf course screen with warnings for the Maintenance Technician user profile; (**d**) Historic games record for the Player profile.

**Figure 18 sensors-23-00047-f018:**
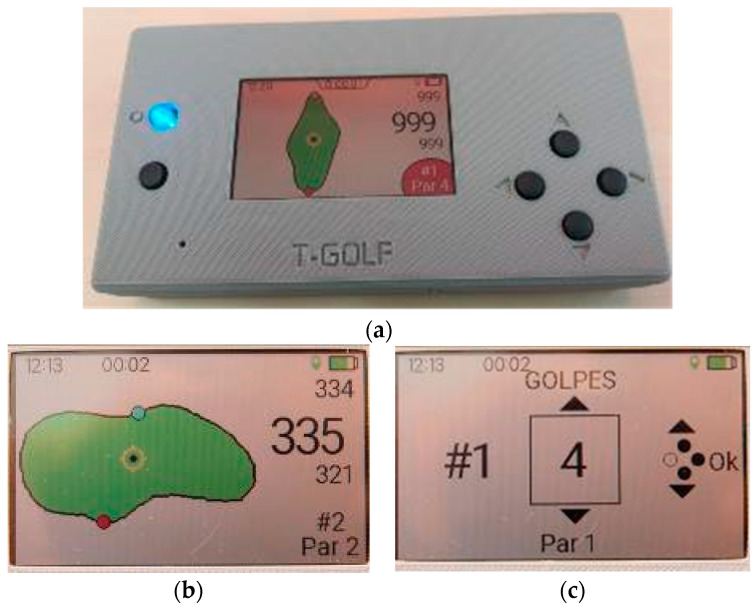
(**a**) USER-ED showing the map of a hole; (**b**) Screenshot of a hole map with info regarding distance to the green, hole number and PAR of the hole, among others; (**c**) Screenshot of the application to introduce the number of made strokes.

**Figure 19 sensors-23-00047-f019:**
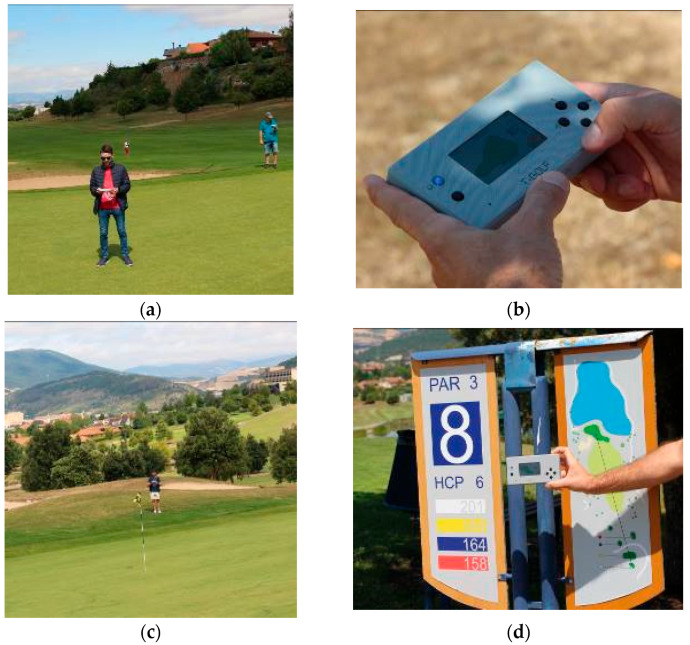
Pictures taken during the test under real conditions: (**a**) The golf player with the USED-ED device; (**b**) The USER-ED device; (**c**) The Golf course during the game; (**d**) The USER-ED device at the beginning of Hole 3.

**Figure 20 sensors-23-00047-f020:**
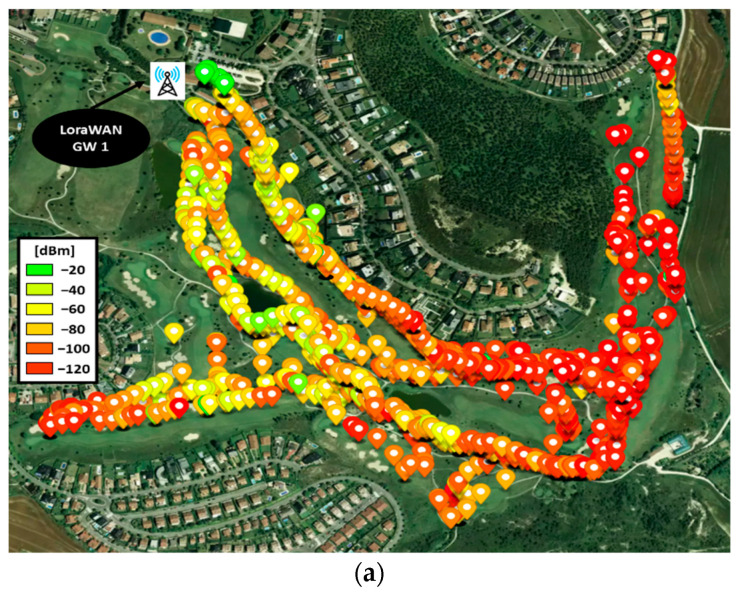
RF results for the test under real conditions: (**a**) Coverage for Gateway 1, located in front of the Club House; (**b**) coverage for the extra Gateway 2; (**c**) coverage for the combination of the two gateways; (**d**) location tracking for the 4 golf players.

**Figure 21 sensors-23-00047-f021:**
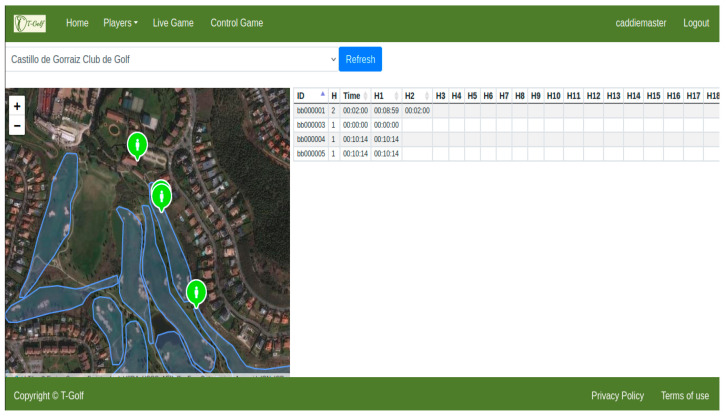
Real-time monitoring of the rounds and localization of players during the test under real conditions.

**Table 1 sensors-23-00047-t001:** Related work concerning golf sport.

Ref	Description	Employed Technologies and Connectivity	Golf Course Management or Monitoring	Golf Game Dynamics Management	Main Contribution
[13]	Measurement of Angular Motion in Golf Swing by a Local Sensor at the Grip End of a Golf Club.	3D accelerometer, 3D gyro sensor.	No	No	Angle and angular velocity measurement of the golf club grip end to train golf players using quantitative data.
[14]	Long Range Battery-Less PV-Powered RFID Tag Sensors.	RFID (800–1000 MHz).	Yes	No	Embedded tracker with PV-RFID tag to find lost golf balls by increasing the distance of range for a few meters.
[15]	Golf Swing Segmentation from a Single IMU Using Machine Learning.	-	No	No	Estimation and division of golf swing phases using kinematic IMU data, eliminating the limitation of the sensor location.
[16]	The Combined Use of Remote Sensing and Wireless Sensor Network to Estimate Soil Moisture in Golf Course.	Soil moisture sensors, GPS device. Wireless connectivity to a hub.	Yes	No	The combined use of remote sensing (using Copernicus Sentinel-2 mission images) and a soil moisture sensor network for maximizing water efficiency.
[17]	Early Improper Motion Detection in Golf Swings Using Wearable Motion Sensors: The First Approach.	3D gyroscope and accelerometer motion sensors.	No	No	Analysis of a golf swing to detect an improper movement in the initial phase of the swing.
[18]	A sensor-aided self coaching model for uncocking Improvement in golf swing.	3D accelerometer, magnetometer and gyro sensors. Microsoft Kinect camera. Bluetooth.	No	No	Wrist angle change analysis during the swing movement which provides 3D rotation data, using two IMU sensors attached to the forearm and the golf club.
[19]	Golf Swing Motion Tracking Using Inertial Sensors and a Stereo Camera.	3D accelerometer and gyro sensors. Eight infrared LEDs captured by a USB cameras.	No	No	Golf club monitoring with an inertial navigation algorithm instead of only estimating the golf club tilt and position.
[20]	Electromyographic Patterns during Golf Swing: Activation Sequence Profiling and Prediction of Shot Effectiveness.	Polhemus Liberty electromagnetic motion capture system at 120 Hz.	No	No	Muscle activity analysis during the golf swing extracting information from electromyographic (EMG) signal stream dynamics to predict the best shot.
[21]	Analysis of swing tempo rhythm, and functional swing plane slope in golf with a wearable inertial measurement unit sensor.	3D accelerometer and gyroscope sensors. Optical motion camera system. Bluetooth.	No	No	Comparison of a swing motion algorithm against an optical motion camera system by estimating the golf club trajectories.
[22]	Three Dimensional Upper Limb Joint Kinetics of a Golf Swing with Measured Internal Grip Force.	A 6-axis force-torque sensor. Infrared cameras.	No	No	A sensor-embedded club developed for measuring the internal grip force and torque during a golf swing.
[23]	Reliability and Validity of the Polhemus Liberty System for Upper Body Segment and Joint Angular Kinematics of Elite Golfers.	-	No	No	Validation of the Polhemus Liberty system for different body parts and angular kinematics at key events during the golf swing.
[24]	Test–Retest Reliability of Task Performance for Golf Swings of Medium to High-Handicap Players.	3D Doppler tracking golf radar.	No	No	Analysis of the test–retest reliability of swing motion variables for medium- to high-handicap players.
[25]	AI Golf: Golf Swing Analysis Tool for Self-Training.	-	No	No	Application for easy correction of the user of the swing motion with image frames and human motion visualization.
This work	Design, Assessment and Deployment of an Efficient Golf Game Dynamics Management System based on Flexible Wire-less Technologies.	LPWAN: LoraWAN, ZigBee. WPT system. Wireless connectivity to a proprietary server.	Yes	Yes	Design, implementation and testing of an optimal and flexible communications system to manage the dynamics of the game and provide precise information to users and managers at any golf course.

**Table 2 sensors-23-00047-t002:** Parameter configuration for the simulations using the 3D-RL tool.

Parameter	Value	
Operation frequency	975 MHz	2.4 GHz
Output power	9 dBm	7.5 dBm
Antenna Gain	−16 dBi	−6 dBi
Height of transmitter and receiver points	1.1 m (over the grass)	
Horizontal ray resolution (∆Φ)	0.5°	
Vertical ray resolution (∆θ)	0.5°	
Max. number of reflections	6	
Mesh resolution	50 cm × 50 cm × 50 cm	

**Table 3 sensors-23-00047-t003:** Overview of LoRaWAN and ZigBee technologies.

Parameter	LoRaWAN	ZigBee
Frequency ranges	EU863-870MHz/EU433 MHz/US902-928 MHz/CN470-510 MHz/CN779-787 MHz/AU915-928 MHz/AS923 MHz	2.4 GHz (ISM) 868 MHz (EU) 915 MHz (US)
Modulation	LoRa modulation/CSS	OQPSK
Transmitted power	EU: 14 dBm US: 20 dBm	10 dBm/18 dBm (max.)
Bandwidth	EU: 125 KHz/250 KHz US: 125 KHz/500 KHz	2 MHz
Data rate	50 Kbps	20 kbps (EU) 40 kbps (US) 250 kbps (ISM)
Sensitivity	−148 dBm	−105 dBm
Messages/Day	EU: UL: airtime of 30 s /DL: 10 messages US: Unlimited	Unlimited
Coverage	Urban: 5 km Rural: 20 km	300 m (LoS) 75–100 m (indoor)
Network topology	Star	Star/Mesh
Battery life	Very high	Medium/High
Security	AES 128 bits	AES encryption standard

**Table 4 sensors-23-00047-t004:** Unique packets received by each ZigBee router and gateway.

ZigBee Devices	Number of Packets
Router 1	215
Router 2	80
Coordinator	2091

**Table 5 sensors-23-00047-t005:** Results of tested encapsulation materials for the WPT system.

Material	Primary vs. Secondary Coil Distance = 5 mm
Polylactic Acid (PLA)	Ok
Acrylonitrile Butadiene Styrene (ABS)	Ok
Polyvinyl chloride (PVC)	Ok
Polycarbonate (PC)	X
Methacrylate	X
Foam	X
Oak panel	Ok
Composite	X
Metal	X

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
