# Peer review of "Design, Assessment and Deployment of an Efficient Golf Game Dynamics Management System Based on Flexible Wireless Technologies"

_sensors, 2022, doi:10.3390/s23010047_

Round 1

Reviewer 1 Report

Work in general is good but I have a set of reservations

1. The figures and pictures in general have very poor accuracy and it is not useful for them to remain in this form and should be improved, but some of them are not readable at all (example Fig. 17).

2. The colors used in Figure 16 are harmful to the reader's eyesight. Inappropriate and unprofessional

3. title number three is duplicated.

4.. The English language needs to be modified and improved

5. The number of references is insufficient and similar case studies should be developed.

======

In terms of content, the work is good, but according to my understanding, it was tested in a real environment in one player.

The question that comes to mind is what are the obstacles in the case of more than one player and what is the cost?

Author Response

Response to Reviewer 1 Comments

Work in general is good but I have a set of reservations.

Point 1: The figures and pictures in general have very poor accuracy and it is not useful for them to remain in this form and should be improved, but some of them are not readable at all (example Fig. 17).

We thank the reviewer's comment and fully agree that there are some figures not readable.

We updated the manuscript by modifying the size and position on the manuscript of most of the figures and pictures and specifically replacing Figures 17 and 21 in order to make them readable, following reviewer suggestion.

Point 2: The colors used in Figure 16 are harmful to the reader's eyesight. Inappropriate and unprofessional

We thank the reviewer's comment and fully agree that Figure 16 is improvable in order to make it more appropriate for the readers.

Therefore, we updated the manuscript by modifying Figure 16 using neutral colors in the background of the boxes and in the main titles.

Point 3: title number three is duplicated.

We thank the reviewer's comment and indeed title number three is duplicated.

We updated the manuscript by changing title number three “Description of the Golf Course” to “RF Propagation within Golf Environments”.

Point 4: The English language needs to be modified and improved.

We thank the reviewer's comment. We have reviewed the use of English language in the full text of the revised manuscript version, following reviewer suggestion.

Point 5: The number of references is insufficient and similar case studies should be developed.

We thank the reviewer's comment. We fully agree that more references are needed in order to compare our work.

We updated the manuscript by adding a benchmark table, named table 1, of related work concerning golf sport. This table summarizes some relevant contributions [13-25] related to golf focused on improving the motion of golf players, specifically the swing movement. However, none of these manuscripts considers the golf course, its monitoring and the dynamics of the game as our work does, which allows an improvement of the playing experience throughout the golf course. This information has been explicitly stated in the revised manuscript version, following reviewer suggestion.

Point 6: In terms of content, the work is good, but according to my understanding, it was tested in a real environment in one player. The question that comes to mind is what are the obstacles in the case of more than one player and what is the cost?

We thank the reviewer comment. This issue, pointed out by the reviewer is a key issue for the real deployment and use of the proposed system. Thus, the authors considered it in the design and the validation activities performed during the test. In the same way, the previous version of the manuscript presented, in Section 4.3, the validation of the proposed system for 4 different golf players, being on the golf course at the same time. For example, Figure 20d presents the tracking data of those 4 golf players. Regarding the obstacles and limits in terms of number of users, the proposed solution is limited by the employed wireless technology itself. That means that for a LoRaWAN solution, a single 8-channel gateway can support a few hundred thousand messages over the course of a 24-hour period. If each node sends, for example, 100 messages a day, such a gateway can support about 1,000 devices. In case ZigBee technology was chosen, the maximum number of devices/nodes connected to that wireless network will be 65,535 (including USER-ED and potential sensors deployed on the golf course for other applications such as monitoring of the state of the greens). Thus, following reviewer suggestions, new text has been included in the new version of the manuscript, stating clearly the number of golf players that were tested at the same time, as well as the information regarding the limits of the proposed system.

Reviewer 2 Report

The submitted manuscript proposes golf management system using existing wireless technologies. The manuscript is written well (minor grammatical mistakes) and the outcomes are presented well. Also, the results shows that the system works well.

However, few queries needed to be answered,

1. The reviewer felt difficulties to find out what is the research gap behind which is needed to be specified very clearly. It means, in a very concrete manner the authors need to declare why this research has been conducted. What novelty is claimed by the authors?

Again, a novel/new work can be done but the philosophy associated with a problem needed to be solve, is very important. 

2. The contribution of this research is needed to be specified clearly. A benchmark table of comparisons with the existing works stating the key improvement is necessary to assess the quality of the work and contribution to scientific knowledge.         

Author Response

Response to Reviewer 2 Comments

The submitted manuscript proposes golf management system using existing wireless technologies. The manuscript is written well (minor grammatical mistakes) and the outcomes are presented well. Also, the results shows that the system works well. However, few queries needed to be answered.

Point 1: The reviewer felt difficulties to find out what is the research gap behind which is needed to be specified very clearly. It means, in a very concrete manner the authors need to declare why this research has been conducted. What novelty is claimed by the authors?

We thank the reviewer's comment and fully agree that the novelty and contribution must be clearly highlighted. The work describes the design, implementation and analysis of a distributed wireless sensor network system that enables golf course monitoring and the dynamics of the game, which allows an improvement of the playing experience throughout the golf course. In this sense, the manuscript has been updated in order to include a comprehensive state of the art in relation with the current solutions, which can provide either Golf course management/monitoring or management of Golf game dynamics, but not both functionalities at the same time. A holistic approach has been followed in order to consider the impact and limitations derived from the use of wireless communication links, with deterministic as well as measurement driven wireless channel analysis as well as system level performance. Ad-hoc devices have been implemented, as well as the corresponding information processing platform in order to collect and analyze the information obtained, adapted to the specific requirements derived from the conditions of the Golf course as well as the game dynamics. This information has been explicitly included in the revised manuscript version in order to provide a sound description of the contribution of the work, following reviewer suggestion.

Point 2: A benchmark table of comparisons with the existing works stating the key improvement is necessary to assess the quality of the work and contribution to scientific knowledge.

We thank the reviewer's comment and fully agree that a benchmark table of existing works in golf sport is necessary in order to compare with our work. This table summarizes some relevant contributions [13-25] related to golf focused on improving the motion of golf players, specifically the swing movement. However, none of these manuscripts considers the golf course, its monitoring and the dynamics of the game as our work does, which allows an improvement of the playing experience throughout the golf course.

We updated the manuscript by adding a benchmark table, named table 1, of related work concerning golf sport, following the reviewer’s suggestion.

Reviewer 3 Report

The paper is well structured, although some sections have the wrong names (Sections 2 and 3 have the same name, likewise Sections 3.1 and 3.2). The topic fits with that of the journal and the paper details a worthwhile application of sensor networks. However, there is the feeling that the experiments that were carried out appear lacking in scientific rigor. The comparison between LoRaWAN and ZigBee are provided in the form of Google Earth overlays. I believe the paper should be enriched with more detailed technical information regarding the coverage and peculiarities of each proposed solution. Authors should also elaborate on the necessity of the simulation presented in Section 3.1 -- is this still required when measurements can be carried out? In addition, a few more minor observations:

  • Perhaps Figures 1 and 2 should either occupy the full width of the page, or half, so that the Google Earth photos, as well as the elevation profile photos can be placed next to each other.
  • The bullet-points mentioned in section 4.1 are not hardware functionalities, they are a means to an end. Functionalities mean "what the system can do/achieve", while these detail the HOW the system achieves those things.
  • I would replace "fabricated" with "manufactured", as fabricated also has a negative meaning (of things being invented, or made up)
  • Screenshots in Figure 17 are not really legible; also, the minutiae of the application's UI is not of much interest from a research perspective

Author Response

Response to Reviewer 3 Comments

Point 1: The paper is well structured, although some sections have the wrong names (Sections 2 and 3 have the same name, likewise Sections 3.1 and 3.2).

We thank the reviewer's comment and indeed sections 3 and 3.2 have the wrong names.

We updated the manuscript by changing section 3 “Description of the Golf Course” to “RF Propagation within Golf Environments” and section 3.2 “RF assessment of a single Golf hole” to “RF Assessment of the Entire Golf Course”.

Point 2: The topic fits with that of the journal and the paper details a worthwhile application of sensor networks. However, there is the feeling that the experiments that were carried out appear lacking in scientific rigor.

The comparison between LoRaWAN and ZigBee are provided in the form of Google Earth overlays.

I believe the paper should be enriched with more detailed technical information regarding the coverage and peculiarities of each proposed solution.

We appreciate the reviewer comment. We totally agree with the reviewer that more detailed technical information regarding the RF analysis for the whole golf course will improve the paper. Following reviewer’s suggestion, new text has been included in Section 3.2.1, providing more technical insight regarding the results presented in Figure 13 and Figure 14. In addition, a new table (Table 4) has been included and commented in the new version of the manuscript. On the other hand, the figures based on Google Earth views have been maintained in the new version of the manuscript, since the authors think they provide a clear and easily understandable view of the gateways and routers deployment, as well as the areas of the golf course covered by each of them.

Point 3: Authors should also elaborate on the necessity of the simulation presented in Section 3.1 -- is this still required when measurements can be carried out?

We thank the reviewer comment very much. This is an interesting comment, indeed. As the reviewer noted, the reasons why this approach has been implemented and presented in this study are not explained well enough in the previous version of the manuscript. The objective of the study of a single hole of the golf course carried out in this manuscript, as it was slightly commented in Section 3, is to analyze the RF behavior within this kind of environments (which is not studied in the literature) for a better understanding of its characteristics to help the development and deployment of potential future applications in such scenarios. For instance, the deployment of devices/sensors on the flag or the hole in order to provide an exact localization of them (changing the holes’ locations within the green every 15 days is a common practice), where the information will be directly sent, wirelessly, to the player’s hardware device. Another application could be monitoring sensors (soil moisture, etc.) deployed on the greens which sent data directly to the maintenance staff, via Bluetooth, for example. Although the measurements can be carried out, simulations provide volumetric RF results (i.e. for the whole 3D volume of the scenario), while the measurements provide data for the specific measurement point. Besides, every hole is different and unique, and taking measurements for every hole will be tortuous. Therefore, the analysis presented in this work validates the employed simulation methodology to assess this kind of scenarios, making this approach appropriate to be used for the analysis of every hole, not only of the presented golf course, but also for every golf course’s holes around the world, without the need of taking measurements or being present there. Following reviewer’s suggestion, new text has been included in Section 3 of the new version of the manuscript for clearly explaining this issue.

Point 4: Perhaps Figures 1 and 2 should either occupy the full width of the page, or half, so that the Google Earth photos, as well as the elevation profile photos can be placed next to each other.

We thank the reviewer's comment and suggestion. In order not to make smaller the size of the elevation profile photos, facilitate the reader's eyesight and keep the manuscript structure, we maintain the position of the figures as they are.

Point 5: The bullet-points mentioned in section 4.1 are not hardware functionalities, they are a means to an end. Functionalities mean "what the system can do/achieve", while these detail the HOW the system achieves those things.

We thank the reviewer's comment and fully agree with the correction.

Therefore, we updated the manuscript by modifying the phrase “The HW functionalities desired for the devices can be summarized as” to “The following points summarized the elements desired from the HW side” at the beginning of section 4.1, following the reviewer’s suggestion.

Point 6: I would replace "fabricated" with "manufactured", as fabricated also has a negative meaning (of things being invented, or made up).

We thank the reviewer's comment and fully agree with the correction.

We updated the manuscript by replacing “fabricated” with “manufactured”, following the reviewer’s suggestion.

Point 7: Screenshots in Figure 17 are not really legible; also, the minutiae of the application's UI is not of much interest from a research perspective.

We thank the reviewer's comment and indeed Figure 17 is not readable. We updated the manuscript by replacing Figure 17 and also Figure 21 in order to make them legible.

The inclusion of the details in relation with the application have been included in order to provide a holistic view of the system, for the sake of completeness, as it´s part of the functional implementation. In this sense the application is a valuable element specially in terms of system validation and testing, as it provides the interface both to perform data extraction as well as to validate user experience (both from the player side as well as of from the gold course management side). Moreover, this information is part of the overall system development, providing an overall view in relation with the functionalities and challenges related with it.

Round 2

Reviewer 2 Report

no comments